# Automatic Segmentation of the Jaws Used in Guided Insertion of Orthodontic Mini Implants to Improve Their Stability and Precision

**DOI:** 10.3390/medicina60101660

**Published:** 2024-10-10

**Authors:** Andra Patricia David, Silviu Brad, Laura-Cristina Rusu, Ovidiu Tiberiu David, Christian Samoila, Marius Traian Leretter

**Affiliations:** 1“Victor Babes” University of Medicine and Pharmacy Timisoara, 2 Eftimie Murgu Sq., 300041 Timisoara, Romania; 2Department of Radiology, “Victor Babes” University of Medicine and Pharmacy Timisoara, 2 Eftimie Murgu Sq., 300041 Timisoara, Romania; 3Department of Oral Pathology, Multidisciplinary Center for Research, Evaluation, Diagnosis and Therapies in Oral Medicine, “Victor Babes” University of Medicine and Pharmacy Timisoara, 2 Eftimie Murgu Sq., 300041 Timisoara, Romania; 4Faculty of Physics, West University of Timisoara, 4 Vasile Parvan Blvd., 300223 Timisoara, Romania; 5Department of Functional Sciences, Multidisciplinary Center for Research, Evaluation, Diagnosis and Therapies in Oral Medicine, “Victor Babes” University of Medicine and Pharmacy Timisoara, 2 Eftimie Murgu Sq., 300041 Timisoara, Romania; 6Radiologie CBCT SRL, Iulius Mall, 2 Consiliul Europei Sq., 300627 Timisoara, Romania; 7Orthodontisimus SRL, 28 Episcopiei Str., 310084 Arad, Romania; 8Department of Prosthodontics, Multidisciplinary Center for Research, Evaluation, Diagnosis and Therapies in Oral Medicine, “Victor Babes” University of Medicine and Pharmacy Timisoara, 2 Eftimie Murgu Sq., 300041 Timisoara, Romania; leretter.marius@umft.ro

**Keywords:** automatic segmentation, guided insertion, orthodontics, mini screws, bicortical, mini implants

## Abstract

*Background and Objectives:* With the goal of identifying regions with bicortical bone and avoiding root contact, the present study proposes an innovative technique for the simulation of the insertion of mini orthodontic implants using automatic jaw segmentation. The simulation of mini implants takes place in 3D rendering visualization instead of Multi-Planar Reconstruction (MPR) sections. *Materials and Methods:* The procedure involves utilizing software that automatically segments the jaw, teeth, and implants, ensuring their visibility in 3D rendering images. These segmented files are utilized as study models to determine the optimum location for simulating orthodontic implants, in particular locations characterized by limited distances between the implant and the roots, as well as locations where the bicortical structures are present. *Results:* By using this method, we were able to simulate the insertion of mini implants in the maxilla by applying two cumulative requirements: the implant tip needs to be positioned in a bicortical area, and it needs to be situated more than 0.6 mm away from the neighboring teeth’s roots along all of their axes. Additionally, it is possible to replicate the positioning of the mini implant in order to distalize the molars in the mandible while avoiding the mandibular canal and the path of molar migration. *Conclusions:* The utilization of automated segmentation and visualization techniques in 3D rendering enhances safety measures during the simulation and insertion of orthodontic mini implants, increasing the insertion precision and providing an advantage in the identification of bicortical structures, increasing their stability.

## 1. Introduction

The area of medicine is undergoing a revolution due to the rapid developments in medical imaging. Cone-beam computed tomography (CBCT), intraoral and face scanners, dental 3D printing, and artificial intelligence are all examples of how quickly digital dentistry is developing. In medical image investigations, it is frequently necessary to differentiate or segment objects, organs, or structures from their surrounding background [1]. The analysis of organs and structures in medical images is becoming increasingly significant in the field of diagnosis and in providing guidance for minimally invasive surgical and therapeutic interventions. For orthodontic diagnosis and treatment planning, it is essential to obtain precise segmentation of the jaw and teeth in CBCT scans [2]. Traditional manual or automated thresholding-based methods, incorporated into commercial or open-source 3D software (Blue Sky Plan software version 4.12.13 (64-bit) (Blue Sky Bio LLC, Libertyville, IL, USA). applications, primarily drive the segmentation process that models these 3D surfaces [3]. In order to facilitate its practical application, the segmentation process should be automated by software. One advantage of segmentation is its ability to generate 3D visible structures, facilitating the verification of the implant simulation’s site through the identification of its spatial surroundings. Micro-implants (OMIs), which are also called temporary anchorage devices (TADs), mini implants, or mini screws in the field of orthodontics, have been used to make complicated orthodontic movements possible [4]. Numerous orthodontic operations that were previously used to manage anchoring have been made simpler by the orthodontic mini screws that give a skeletal anchorage, and the side effects of many orthodontic appliances have been thus minimized [5]. Although these mini screws were originally inserted manually, thanks to the benefits of digitization, especially in particularly difficult cases, guided insertion is now used [6]. Previous studies have assessed the precision of computer-guided mini implant insertion [7,8,9]. Bae et al., in a study using cadaver jaws, discovered that 20% of the direct implant insertions resulted in contact with the roots; in contrast, there were no incidences of this in the guided implant placement group, as shown in Figure 1 [10].

The accuracy results show that, in comparison to implants inserted manually, implants inserted with guidance systems offer significantly better precision, as reported in computer-guided implantology studies [11,12]. Even in the hands of less skilled medical professionals, CAD/CAM templates enable more accuracy and consistency in the insertion of mini screws, as well as improved orientation and depth of the cortical insertion. This can lead to a shorter recovery period following surgery and a lower chance of damage to adjacent anatomical tissues, all of which can improve patient comfort [13,14]. While the use of surgical guides contributes to the accuracy of mini implant placement, the stability of these implants also depends on a qualitative examination of the bone structures present at the insertion site, including the identification of bicortical and tricortical structures situated where the vestibular wall of the jaw and the maxillary sinus floor meet or where the nasal floor meets the maxillary sinus floor [15]. Additionally, for maxillary distalization, the infrazygomatic ridge area is the place where the mini implants are placed, particularly in cases of class II malocclusion [16]. The spaces in the jaw between the two premolars and the space between the second premolar and the first molar are the most often used locations for mini implant insertion [17,18]. When considering the utilization of mini implants in the mandible for the purpose of distalizing the molars, it has been determined that the most advantageous location for implantation is situated medially and distally to the second molar [19,20].

This study aims to examine three cases, two in the maxilla and one in the mandible, with the help of automated segmentation. A study model is utilized to simulate the mini implant with the help of in space visibility using 3D rendering, offering an exhaustive examination of the interactions between the mini implant and the surrounding structures. In the first two cases, we will examine the maxilla in the mesial and distal regions of the second premolar. The mesial position relative to the second premolar anatomically corresponds to the location of the intersection between the nasal cortical and the palatal cortical at the mini implant’s tip. In this particular case, our objective is to determine the region of intersection between the two cortical structures in order to properly situate the implant in that location, which improves its stability. The position distal to the second premolar corresponds to the intersection between the maxillary sinus cortical and the maxillary vestibular cortical at the mini implant’s tip. Additionally, in this case, we will look for the point where the two corticals connect, to provide stability. Additionally, we will look for a safety zone where the mini implant passes between the two roots of the corresponding neighboring teeth, at least one millimeter from each one [21]. In the mesial zone of the second molar, the third case will be located on the mandibular buccal shelf [22], as this insertion place was determined to be the most suitable for the mandible [18].

A variety of software applications to automate segmentation have been developed [23]. Specifically for the dental field, we identified two automatic segmentation software applications: BlueSkyPlan version 4.12.13 (64-bit) (Blue Sky Bio LLC, Libertyville, IL, USA) [24] and Diagnocat (Diagnocat Inc., San Francisco, CA, USA) [25]. In contrast to the second, which requires payment for segmentation, we have utilized the first one, since it provides automatic segmentation for free and is open source [26].

An inherent constraint of the existing technique employed for simulating the placement of orthodontic mini implants is its capability to conduct this simulation exclusively within the sections of the 3D image in Multi-Planar Reconstruction (MPR) visualization, instead of in the spatial domain within the 3D rendering interface. The number of sections in the imaging software is limited. Typically, the mentioned orientations include the sagittal, coronal, axial, cross-sectional, and panoramic orientations. If the implant is simulated in certain previous sections, its orientation will be limited due to the absence of oblique planes, where the mini implant may be rotated in these sections. Only the possibility of its spatial orientation in all three planes gives it the security of insertion where it is desired and where it meets the conditions of stability and precision.

The practical method proposed in this study solves this issue by providing the potential for unrestricted spatial orientation of the mini implant.

## 2. Materials and Methods

We conducted the present study using the Blue Sky Plan software version 4.12.13 (64-bit) (Blue Sky Bio LLC, Libertyville, IL, USA). The workflow diagram is shown in Figure 2.

We first selected the Model Master module. Then, we selected the “Import Patient CT Scan” option and imported the patient’s DICOM image into the application. The DICOM 3D files utilized in this work have a field of view of 5 × 8 cm, a slice thickness of 0.2 mm, and voxel dimensions of 0.2 × 0.2 × 0.2 mm. The acquisition was performed using an X-ray tube current of 8 mA and a KVP of 89, utilizing the PaX-i3D model from Vatech Company Ltd. (Hwaseong, Republic of Korea). The software manufacturers do not supply the grayscale threshold values utilized for segmentation. These are not adjustable to different values during the start of the automatic segmentation command. If we believed that aligning the datasets was essential to obtain an occlusal plane closer to the horizontal for the reslice function, we did so. Then, we selected the “Automatic Jaw Segmentation” feature from the Tools menu after closing the Wizard that opens in the subsequent window; next, selected the mandible or maxilla to be segmented; and then clicked the button “Start Automatic Segmentation”. The operating system setting determines how long segmentation takes, but it usually takes two to three minutes. We clicked the “Create surface” button once the automatic segmentation was complete. The quality of segmentation in the MPR sections can be checked by modifying the segmented image using the inflate and deflate functions provided by the software by adding or removing a voxel on the entire segmented contour. We navigated to the Advanced tab from the Model Master tab to simulate mini implants, and selected the type of mini implant from the Implant Library or used a custom generic implant that has the same dimensions as the original model in case the corresponding model was not available. The dimensions of the customized implant can be as close to those of the real implant as possible, with a cylindrical or conical forms. Generic implants, created with software slightly larger than original implants, offer increased safety at the insertion site. One limitation is that the lateral spaces between the roots and the mini implant are on the order of tenths of millimeters, and when the mini implants are simulated, it is frequently required to know the real dimensions of these spaces. Thus, compared to generic customized implants, the original implants from the Implant Library are more appropriate for simulating implants. Once the implant has been positioned at the desired place in 3D rendering, the jaw can be hidden, and only the teeth and the implant become visible in 3D renderings, thanks to the separate teeth viewing feature. The software’s placement capability allows the implant to move freely in three dimensions in any direction, allowing for the selection of the most advantageous position in relation to the available spaces and distances from nearby roots. The implant simulation can be corrected from the 3D rendering window in cases where there is an excessively tight closeness between the implant and the tooth roots, thanks to the 3D imaging of the segmented structures. It is more difficult to visualize the spaces between the roots and the mini implant in the MPR view because of the varied angles of the root axes. These spaces should not be, ideally, smaller than 0.6 mm, to avoid contact with neighboring roots [27].

## 3. Results

From the CBCT DICOM file, Figure 3a, we can obtain unique STL files for each tooth, as well as for the mandible and maxilla, or for the mandibular canal, by using automated segmentation in BlueSkyPlan software (Blue Sky Bio LLC., Libertyville, IL, USA). They can be studied either together or separately (Figure 3a–d).

In the first case of this study, we could locate the ideal position for the simulated mini implant between the two premolars, which was equally spaced from the surrounding roots, and acquire the bicortical structure made up of the sinus maxillary floor and the nasal floor. The buccal side, the palatal side of the mini implant, and the two premolars are shown Figure 4a,b, respectively. In Figure 4c, the STL file of the maxilla is sectioned with a plane that contains the mini implant’s axis to verify the image of the biortical position. Figure 4d provides evidence supporting the positioning of the mini implant tip within the bicortical area, in the cross-sectional section. The mini implant’s angle of 57 degrees with the horizontal is within the range studied in the article by Wang et al. [19].

In the second investigated case, which refers to the placement of a mini implant between the first molar and the second premolar, the circumstances correspond with those of the initial case in relation to the situations observed during the implant simulation. Figure 5a displays a section that crosses the axis of the implant, revealing the distance from the neighboring roots exceeding the threshold of 0.6 mm. Figure 5b provides illustrations of the implant’s movement in three-dimensional space during his movement to determine the ideal simulation position. In Figure 5c, the maxilla STL file is sectioned with a plane containing the mini implant’s axis to confirm the rendering of the bicortical position. Figure 5d demonstrates that the mini implant tip is positioned within the bicortical area in the cross-sectional section. The mini implant’s angle of 45 degrees with the horizontal is within the range analyzed in the publication by Wang et al. [19].

In the third case, with the insertion of the mini implant between the first and second mandibular molars, the aim was to avoid the mandibular canal and simulate the mini implant, keeping away from the path of movement of the molars towards distalization (Figure 6).

It is important to show caution when using automatic segmentation by verifying the boundaries of the jaw bone and the outside boundaries of the teeth in the MPR software sections in order to ensure that the segmented contours closely correspond to the real dimensions observed solely within the sections. Figure 7 illustrates the segmentation of the same mandible using two distinct software applications. The segmentation in Figure 7a was performed using BlueSkyPlan software (Blue Sky Bio LLC, Libertyville, IL, USA), while the segmentation in Figure 7b was performed using Diagnocat software (Diagnocat Inc., San Francisco, CA, USA). We imported the STL file of the segmented mandible from Diagnocat (Diagnocat Inc., San Francisco, CA, USA) into the Blue Sky Plan software (Blue Sky Bio LLC, Libertyville, IL, USA) to achieve overlap of the two segmented mandibles. We successfully completed the desired match by utilizing the automated overlay function of the Blue Sky Plan software (Blue Sky Bio LLC, Libertyville, IL, USA), as shown in Figure 7c. The cross-sectional section presented in Figure 7d reveals that there are no major differences seen in the cortical area. The distinctions between them are evident only at the alveolar process, where the vestibular bone shows a notably thin structure. The boundaries of the first segmentation made in the BlueSkyPlan (Blue Sky Bio LLC, Libertyville, IL, USA) are highlighted in orange in the cross-sectional section, while the second segmentation from the Diagnocat (Diagnocat Inc., San Francisco, CA, USA) is highlighted in green.

## 4. Discussion

Many techniques to semi-automatically segment different anatomic structures in CBCT scans have been proposed in the past few decades [2]. The constantly progressing software providing automatic segmentation addresses the necessity for mini implant insertion precision in very small locations. Among these automated techniques are statistical shape models, morphologic snakes, random forests, region seeding, edge detection, and watershed segmentation. In the research paper by Wang et al., they stated that there is still a lack of fully automated segmentation methods that can simultaneously segment both anatomic structures in CBCT scans (i.e., multiclass segmentation) [2]. A lot of research has been conducted using convolutional neural network segmentation [28,29,30], manual segmentation [31,32], or semi-automatic segmentation [33,34] from CBCT images. The process of segmentation, other than automatic methods, is time-consuming, necessitates the use of complex applications and a high level of knowledge for its operation, and is also expensive [31]. These images can be simply examined in terms of their proximity to neighboring structures or their connections with other anatomical processes. It is much harder to see these spaces between the roots and the mini implant in the MPR view because of the varied angles of the root axes. The necessity to improve the precision of simulating mini implant insertion, particularly in small spaces, has led to research efforts focused on their insertion utilizing virtual navigation systems. The utilization of augmented reality technology in navigation procedures has been found to have a significant impact on the precision of orthodontic self-drilling mini implant placement, leading to a reduction in intraoperative problems when compared to the traditional free-hand technique [35,36]. Contact between the mini screws and the tooth root next to the insertion site was one of the most frequently reported problems [37,38,39]. The quantity and quality of bone, as well as the location of the mini implants in the bone, all affect the manner in which the primary stability of the implants occurs. This stability can be maximized by considering the thickness and properties of the bone when choosing the implantation site [40,41,42]. In studies about inserting mini implants in the palate, artificial intelligence and automatic segmentation have been used [24,43].

The differences among various techniques employed by the operators for the insertion of orthodontic mini implants relates to the utilization of the conventional freehand approach versus the guided procedure that incorporates surgical guidance. Three-dimensional images produced by automatic segmentation can be rapidly and simply used as study models.

This article names only two software programs that perform automatic segmentation due to a lack of information on others, particularly those that offer free and user-friendly automatic segmentation for doctors.

The BlueskyPlan software (Blue Sky Bio LLC, Libertyville, IL, USA) is an open-source application combining multiple modules for the manufacturing of orthodontic aligners, surgical guides, cephalometric analyses, and dental crowns and bridges. It remains free until the user chooses to export a set of aligners or surgical guides. The Diagnocat software (Diagnocat Inc., San Francisco, CA, USA) operates on a platform that allows users to upload a 3D DICOM dataset, which then performs automatic segmentation based on the selected criteria: maxilla, mandible, teeth, etc. The Diagnocat software is not free; a fee is charged for each segmentation.

Most software options require advanced computer skills for effective automatic segmentation.

The segmentation method used for this study is reproducible, as it utilizes an algorithm from automated segmentation software that necessitates no supplementary configurations for running. The automatic segmentation operates using default commands and non-configurable commands, resulting in identical segmentation outcomes. The method is applicable in everyday situations; however it necessitates a medium-term learning curve concerning the movement of implants, bones, or teeth as objects in augmented reality. The method is accessible to anyone at no cost.

This article only allowed for a presentation of the concept. Among the shortcomings of this method are the following: a medium-term learning curve for operators unfamiliar with handling 3D rendering volumes, the necessity of equipping computers with processors capable of performing segmentation in a relatively brief period, and the requirement for full-capacity operation of resource-intensive software dependent on computer specifications.

## 5. Conclusions


1.Utilizing open-source software that offers free automatic segmentation is quite helpful for studying 3D CBCT DICOM images.2.Placing an implant in a 3D rendering view significantly decreases the limitations associated with simulating its insertion within the software’s available section plans.3.Using this method, the implant and the tooth may be studied as two three-dimensional objects whose positions can be changed in accordance with different needs.4.When comparing the evaluation of the simulation of implants in 3D rendering images and their visualization in MPR to the three examples provided in this research, the volumetric visualization method provided here is superior.


The automatic segmentation presented in this study topic is useful for those who use software for creating surgical guides for the insertion of orthodontic mini implants, as well as for medical professionals who wish to use the 3D segmentation images as a study model to create an appropriate treatment plan. The automatic segmentation performed by the computer using the software is substantially preferable than semi-automatic or manual segmentations regarding the time resources available to the physician during medical procedures.

## Figures and Tables

**Figure 1 medicina-60-01660-f001:**
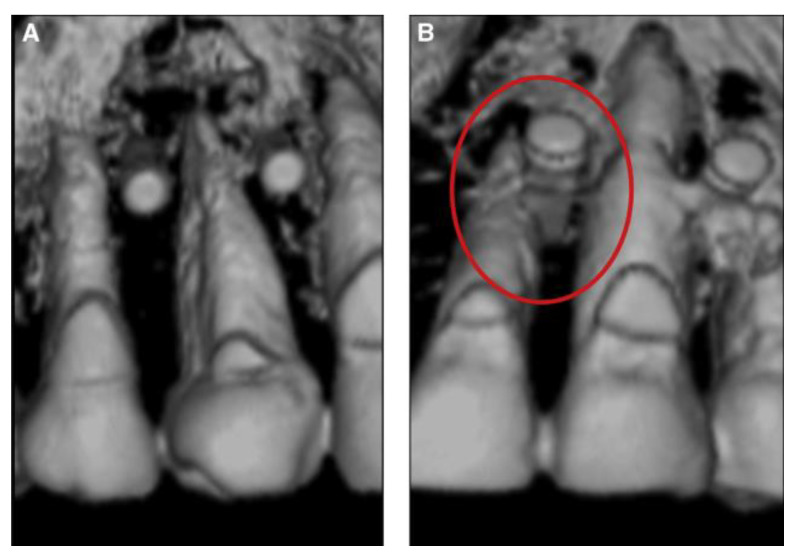
Mini implants inserted for anchoring purposes: (**A**) correctly placed mini implants using a guidance system; (**B**) mini implants inserted incorrectly, without a guidance system, in contact with the root [10] (reproduced with the publisher’s permission using RightsLink licensing number 5734140903513).

**Figure 2 medicina-60-01660-f002:**
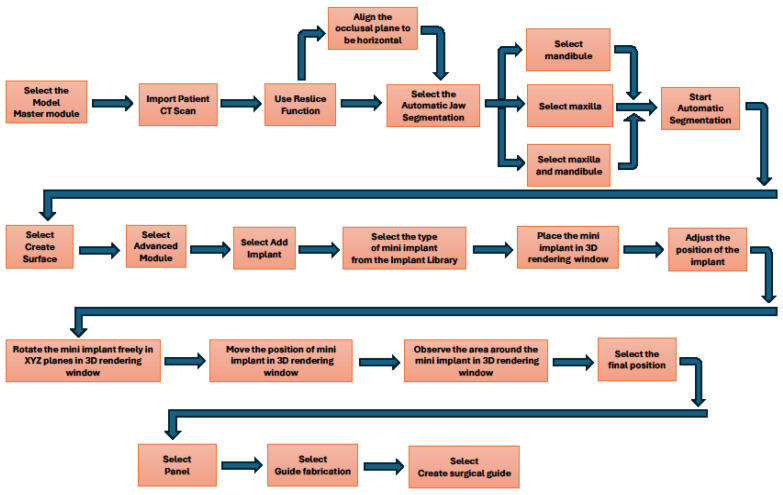
The workflow diagram.

**Figure 3 medicina-60-01660-f003:**
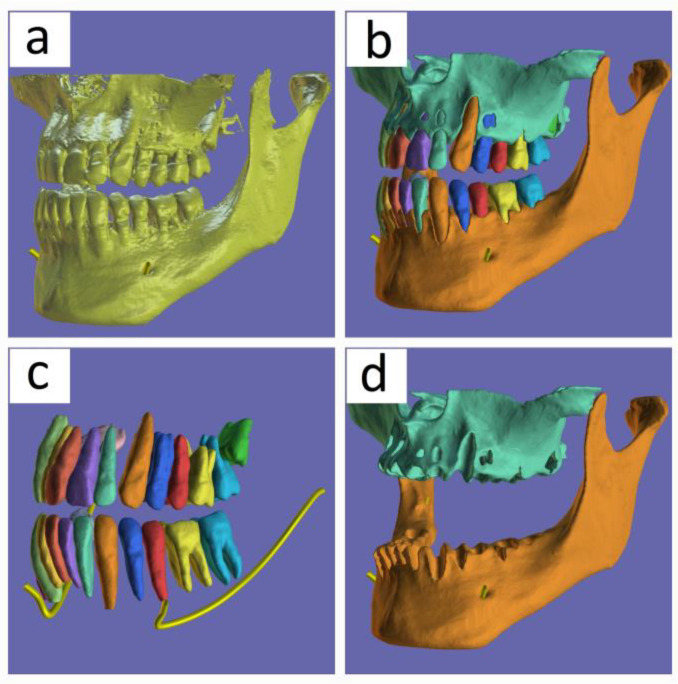
CBCT scan and automatic segmentations of the jaws and teeth: (**a**) 3D rendering view of the DICOM file from the CBCT scan; (**b**) the result of the automatic segmentation of the mandible, maxilla, and teeth being viewed together, with each jaw and all of the teeth having a distinct STL file; (**c**) a separate view of segmented teeth and mandibular canals; (**d**) a separate view of segmented jaws.

**Figure 4 medicina-60-01660-f004:**
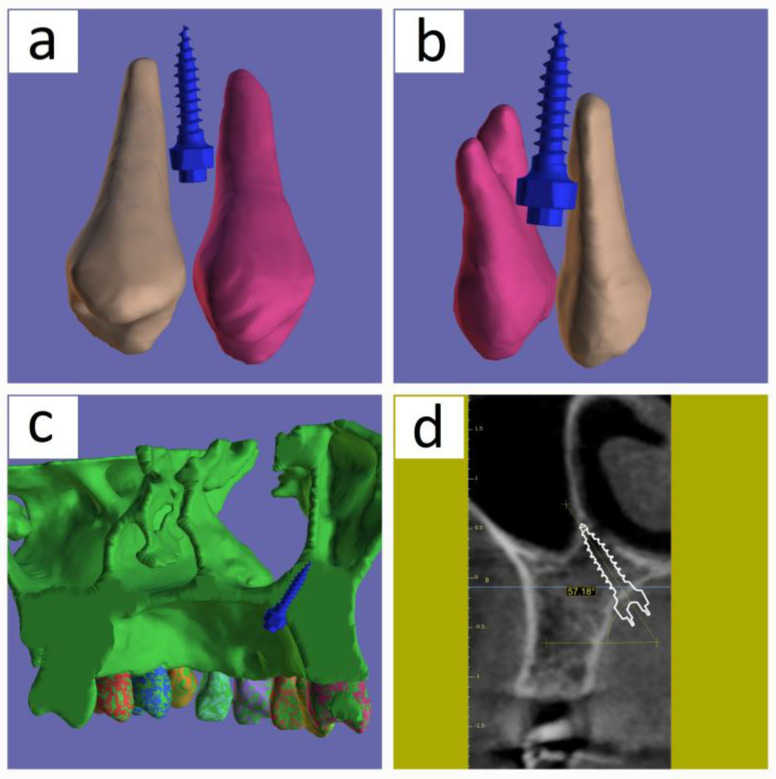
The position of the mini implant simulated in 3D rendering located between the two preolars on the mandible: (**a**) a space view of the two premolars taken from the buccal side, showing the implant located in between them; (**b**) a space view of the two premolars taken from the palatal side, showing the implant located in between them; (**c**) a plane that goes through the mini implant’s axis is used to section the segmented STL file of the jaw. The view is from the posterior side and shows the tip of the implant, where the maxillary sinus floor and the nasal floor cortical connect each other; (**d**) a section taken through the mini implant’s axis in the MPR that also displays the implant’s 57 degrees angle with respect to the horizontal.

**Figure 5 medicina-60-01660-f005:**
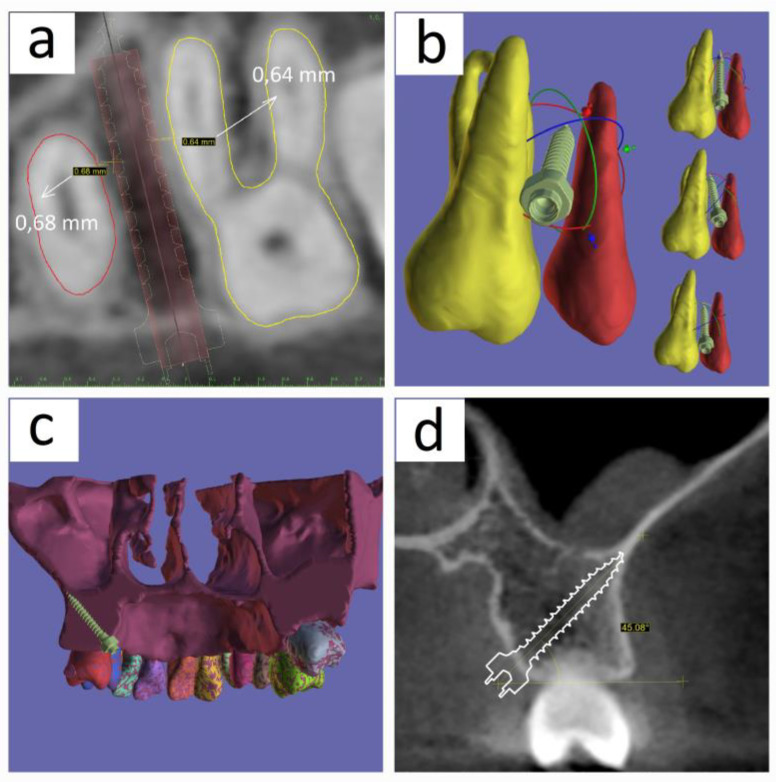
The position of the mini implant simulated in space between the first molar and second premolar on the mandible: (**a**) a space view of the two teeth taken from the buccal side, showing the implant located in between them.; (**b**) a space view of the two teeth taken from the palatal side shows the implant located in between them; (**c**) a plane that goes through the mini implant’s axis is used to section the segmented STL file of the jaw. The view is from the posterior side and shows the tip of the implant, where the maxillary sinus floor and the vestibular cortical of the maxillary sinus connect each other; (**d**) a section taken through the mini implant’s axis in the MPR that also displays the implant’s 45 degrees angle with respect to the horizontal.

**Figure 6 medicina-60-01660-f006:**
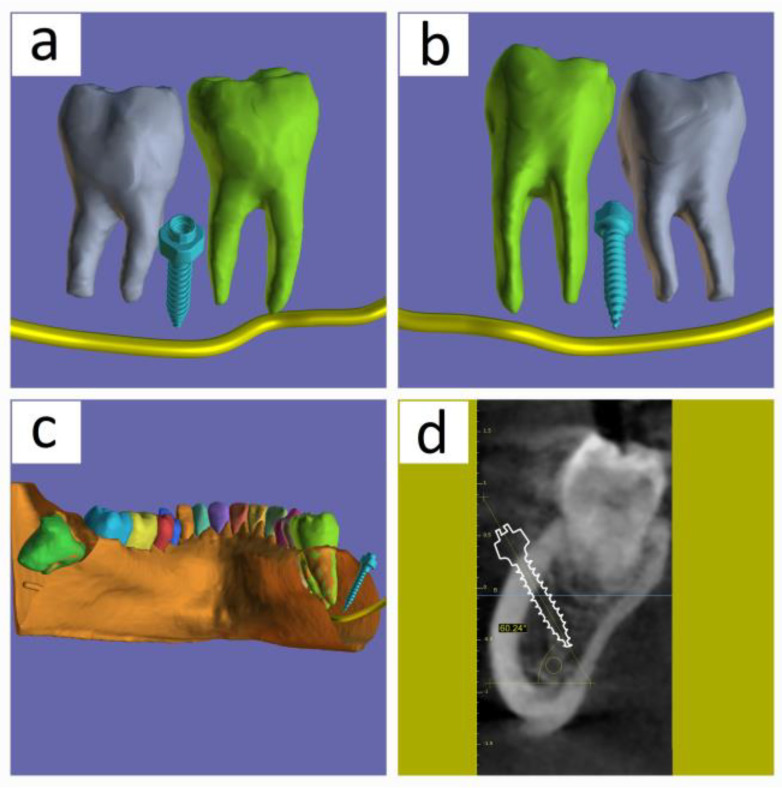
The position of the mini implant simulated in the 3D rendering window between the first molar and second molar on the mandible: (**a**) a space view of the two teeth taken from the buccal side, showing the implant located in between them and its relationship to the mandibular canal; (**b**) a space view of the two teeth taken from the palatal side shows the implant located in between them and its relationship to the mandibular canal; (**c**) a plane that goes through the mini implant’s axis is used to section the segmented STL file of the mandible. The view is from the posterior side and shows the tip of the implant situated vestibular to the teeth’s roots and the mandibular canal, avoiding crossing the path where the teeth are moved during orthodontic treatment; (**d**) a section taken through the mini implant’s axis in the MPR that also displays the implant’s 60-degree angle with respect to the horizontal.

**Figure 7 medicina-60-01660-f007:**
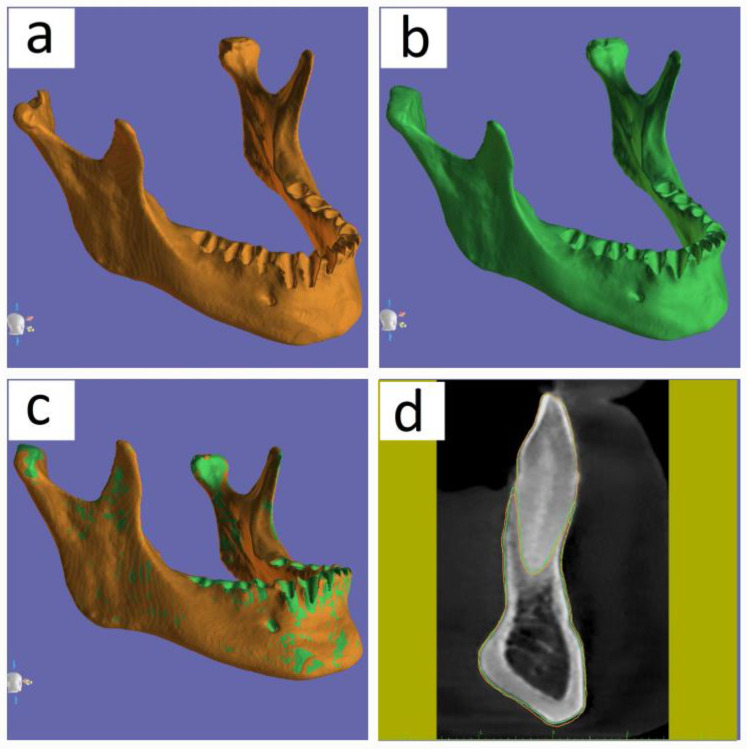
Segmented mandible with two different software: (**a**) mandible segmented with BlueSkyPlan software (Blue Sky Bio LLC, Libertyville, IL, USA); (**b**) mandible segmented with Diagnocat software (Diagnocat Inc., San Francisco, CA, USA); (**c**) right side view of the overlap of the two segmented mandibles with the differences between them; (**d**) cross-sectional view of the overlap of the two segmented mandibles with the differences between them.

## Data Availability

The original contributions presented in this study are included in this article; further inquiries can be directed to the corresponding authors.

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
