# Peer review of "Automatic Segmentation of the Jaws Used in Guided Insertion of Orthodontic Mini Implants to Improve Their Stability and Precision"

_medicina, 2024, doi:10.3390/medicina60101660_

Round 1

Reviewer 1 Report

Comments and Suggestions for Authors

Authors in their work "Automatic segmentation of the jaws used in the guided insertion of orthodontic mini-implants to improve their stability and precision" have put forth a method for positioning the mini-screws by utilizing the 3D reconstruction from the CBCT. They claim that the method proposed by them enhances safety measures during the simulation and insertion of orthodontic mini implants.

Compared with other publication, this method will aid the utilization of augmented reality in navigation procedures, which has a significant impact on the precision of orthodontic self-drilling mini-implant placement, leading to a reduction in intraoperative problems when compared to the traditional free-hand technique.    They have presented the method in detail in a step-by-step manner to accomplish the task by using the recommended software in three cases.    The method seems to be of practical benefit during the planning of mini-screw placement.

The method seems to be of practical benefit during the planning of miniscrew placement. 

Although the authors have postulated an innovative approach, there is a lack of evidence to support that this is a repeatable method that can be done in practical scenarios. Thus, the authors should provide information about inter-operator differences and ease of simulation using the automated segmentation of jaws through statistical analysis to strengthen the usefulness of this method. Authors must also include the shortcomings of this approach.

Overall, this method will aid the utilization of augmented reality in navigation procedures, which has a significant impact on the precision of orthodontic self-drilling mini-implant placement, leading to a reduction in  intraoperative problems when compared to the traditional free-hand technique. Conclusions can be improved by incorporating methods of comparisons among different operators.

Reviewer 2 Report

Comments and Suggestions for Authors

Dear authors, your manuscript presents a clinically relevant topic, however, some changes must be made to improve the content of your manuscript.

1- It would be helpful if you included a workflow diagram showing the progression of the steps involved in the segmentation method used in your study, Right now, what is presented in the materials and method is very detailed, but is hard to follow.

2-Please provide details on the quality of the data used for the segmentation. Details such as voxel size, slice thickness, and the grayscale threshold are used to define the areas to be segmented.

3- I see the 3D volumetric models generated from the CBCT data look smoothened, this can affect the accuracy of the 3D model reconstruction. Please provide details regarding if this is the case.

4-Please extend your discussion by including a brief description of the different segmentation technologies mentioned.

5- Check your images, please make sure the fonts, the size, and the white boxes with the letters have the same dimensions. Right now they look different.

6- Please include a paragraph in your discussion mentioning the limitations of your study. For example, you only used 2 segmentation software.

Round 2

Reviewer 2 Report

Comments and Suggestions for Authors

All my recommendations and comments were properly addressed.